# Tomato By-Products, a Source of Nutrients for the Prevention and Reduction of Malnutrition

**DOI:** 10.3390/nu14142871

**Published:** 2022-07-13

**Authors:** Kabakdé Kaboré, Kiéssoun Konaté, Abdoudramane Sanou, Roger Dakuyo, Hemayoro Sama, Balamoussa Santara, Ella Wendinpuikondo Rakèta Compaoré, Mamoudou Hama Dicko

**Affiliations:** 1Laboratory of Biochemistry, Biotechnology, Food Technology and Nutrition (LABIOTAN), Department of Biochemistry and Microbiology, University Joseph KI-ZERBO, Ouagadougou 03 BP 7021, Burkina Faso; mehekiessoum@yahoo.fr (K.K.); sanouabdoudramane@gmail.com (A.S.); rogerdakuyo@gmail.com (R.D.); hemayorosama@yahoo.com (H.S.); rakibf@gmail.com (E.W.R.C.); mamoudou_dicko2004@yahoo.fr (M.H.D.); 2Laboratory of Biochemistry and Applied Chemistry, Department of Biochemistry and Microbiology, University Joseph KI-ZERBO, Ouagadougou 03 BP 7021, Burkina Faso; 3Applied Sciences and Technologies Training and Research Unit, Department of Biochemistry and Microbiology, University of Dedougou, Dedougou 09 BP 176, Burkina Faso; 4Unité de Formation et de Recherche en Sciences de la Vie et de la Terre, Université Nazi BONI, Bobo-Dioulasso 01 BP 1091, Burkina Faso; santaramoussa13@gmail.com

**Keywords:** tomato, nutritional value, by-product, bioactive compound, phytonutrients, antioxidant activity

## Abstract

Malnutrition is a global phenomenon that mainly affects children under five years old, the elderly and food insecure people. It can be linked to undernourishment or overnutrition. To prevent it, a healthy and balanced diet, rich in energy and micronutrients, is necessary. The objective of this study was to evaluate the nutritional composition and contribution of tomato by-products to prevent and reduce malnutrition. Thus, standard methods were used for proximate composition and bioactive compounds. Results showed that tomato by-products are rich in macronutrients and micronutrients capable of preventing undernutrition and reducing the incidence of the effects of overnutrition. The average carbohydrate, protein and lipid contents were 15.43%, 11.71% and 5.4% (DM) in peels and for seeds the contents were 58.75%, 15.4% and 22.2%, respectively. The average energy values were 280.47 kcal/100 g and 472.8 kcal/100 g DM for peels and seeds, respectively. The main minerals found were, in decreasing order, potassium, magnesium, sodium, iron and zinc. High contents of phenolic compounds, lycopene, β-carotene and vitamin C were also found in these by-products. They also presented important antioxidant activities. Due to their nutritional and bioactive compounds, tomato by-products may be included in functional food formulation programs to reduce the incidence of nutritional diseases.

## 1. Introduction

Malnutrition is a public health problem that affects several countries in the world, developed and developing countries alike [1]. Worldwide, 1.9 billion adults are overweight or obese, while 462 million are underweight. Among children under five years old, 52 million are emaciated, 17 million are severely emaciated and 155 million are stunted, while 41 million are overweight or obese [2]. Undernutrition is involved in approximately 45% of the deaths of children under five years old. These deaths occur primarily in low- and middle-income countries [3]. This form of malnutrition is often associated with stunting, increased risk of severe disease and high mortality. Acute malnutrition occurs in poor countries and in communities living below the poverty line [4]. In recent years, insecurity, combined with the effects of climate change, has exacerbated acute and chronic malnutrition. The main causes of malnutrition, in addition to those mentioned above, are an insufficient supply of nutrients essential to the proper functioning of the body. They are generally linked to low levels of protein and energetic nutrients such as lipids and carbohydrates and also micronutrients [5]. To fight against malnutrition, governments and organizations are working to increase the income of populations. It is also essential to find other sources of nutrients in a context where food is becoming increasingly scarce for the growing demographic. Interestingly, studies have shown that by-products from the food industry are sources of nutrients at low cost if they are accessible. Among these wastes or by-products, tomato wastes have been extensively documented. Being a global production, the access of these tomato by-products in the fight against malnutrition could help organizations and states involved in this fight. Indeed, the tomato is one of the most important world productions. In 2018, the world production was around 182 million tons [6]. This production has increased to a value of over 186 million tons in 2020. About 36.8 million tons of the world’s production are processed fresh, releasing considerable amounts of by-products such as peels and seeds [7]. In Burkina Faso, tomato production in 2020 was estimated to be 21,369 tons [8]. The nutritional value of tomato peels and seeds has been indicated. Many studies have reported that by-products are rich in bioactive compounds including polyphenols, lycopene, β-carotene, vitamin C and dietary fibers. 

Dietary fibers have many biological functions such as reducing the risk of obesity, type-II diabetes, cancer and intestinal diseases [9]. Phenolic compounds are known as natural antioxidants that protect against oxidative stress and chronic diseases. Polyphenols are involved in the treatment of obesity complications and improve insulin resistance and hepatic steatosis. They also have hypoglycemic, anti-inflammatory properties and reduce weight gain [10]. Lycopene and β-carotene are the two most studied carotenoids in tomato by-products. They are antioxidant compounds that protect the body from free radicals. Lycopene prevents cardiovascular diseases and cancer, is involved in the metabolism, limits intestinal damage and reduces the loss of immunoglobulin A [11]. Beta-carotene (provitamin A), in addition to its antioxidant properties, is involved in the regulation of fat and cholesterol metabolism and plays an anti-inflammatory and anticancer role [12]. 

Tomato by-products may be cheap sources of nutritionally interesting minerals to prevent malnutrition due to mineral deficiency [13]. Complex carbohydrates, also present in tomatoes, facilitate intestinal transit and can prevent diabetes mellitus thanks to their low glycemic index [9]. Proteins play the role of building materials. They are essential for growth and the need for them is greater during the first months of life. They serve to continually renew all tissues. Proteins must provide at least 10% of total calories. Malnutrition is often accompanied by protein deficiency, especially serum albumin. Increasing the protein intake can be particularly recommended to fight against this malnutrition. However, this intake must have a balance between essential and non-essential amino acids, as an imbalance negatively influences the digestibility of proteins [14]. Animal proteins are generally associated with an increased risk of cardiovascular disease, kidney disease, insulin resistance and gout. Thus, their consumption should be moderated, especially from red meat, in favor of vegetable proteins. However, the intake of a mixture of several vegetables is recommended [15].

The oil composition of tomato seeds revealed that they are very nutritious and contain a high proportion of unsaturated fatty acids with more than 62% of essential ones. However, this composition varied from one cultivar to another [16]. On the one hand, genotypical difference, year of harvest and soil influence the physicochemical composition of the oil. On the other hand, treatment techniques (hot or cold break) do not impact this composition, which testifies its stability [17]. Because of the affinities between the different molecules of the oils and the organic solvents, the quality and yield of extraction of tomato seed oil differs from one solvent to another [18]. Another nutritional interest of tomato seed oil is its richness in sterols and policosanols (fatty alcohols), the consumption of which is associated with a reduction in LDL cholesterol and prevents inflammation. However, from one oil to another, their contents may differ because they depend on cultivar [19,20]. Foods rich in omega-3 fatty acids protect against the risks of occurrence of cardiovascular diseases and mortality [21]. Unfortunately the stability of omega-3, due to poly-unsaturation, is a major problem for conservation and requires the presence of antioxidants [22]. Of all the benefits of tomato seed oils, its consumption may be encouraged because of its composition in bioactive compounds that have proven to have positive effects on human health [23]. For instance, phytosterols and policosanols present in tomatoes are known to be anti-hypercholesterolemic [24]. Among polycosanols, octacosanol is present in the by-products and also exerts various biological effects, such as anti-fatigue, anti-hypoxia, antioxidant, anti-inflammatory and antitumor effects, and prevents oxidative stress and cardiovascular diseases [25]. The biochemical composition of the plants varies according to the pedoclimatic conditions and the cultivars. Therefore, it is thus important to know the differences in nutritional composition among cultivars and also the effect of cultivation on levels of macronutrients as well as micronutrients. This evaluation of the nutritional composition will contribute to the fight against malnutrition.

## 2. Material and Methods

### 2.1. Material

#### 2.1.1. Vegetable Material and Site of Study

The plant material consisted of seeds and peels of two tomato cultivars (Petomech and F1 Mongal). Samples, from conventional agriculture, were collected in four different regions of Burkina Faso and brought to a laboratory. The study was conducted during three different rainy seasons of 2019, 2020 and 2021. In the study zones, temperatures varied from 19 °C during the cold nights of December to 42 °C during the hot days of June and July. The climate was Sahelo-Sudanese characterized by two distinct seasons: a long dry season from October to May and a short rainy season from June to October. The soils had low organic matter content, and were bare and heavily degraded due to the overexploitation of natural resources, soils and vegetation cover. Thus, the fertilization systems used were NPK and organic manure. Drip irrigation was used during the dry season. The production was conducted according to technical manuals where the quantity of water used varied from 15 mm to 20 mm per week for the cold times. For the hot periods of the dry season, the quantities of water varied from 80 mm to 90 mm per week. The soils had a pH between 6.8 and 7.2 and were better adapted to the cultivars used due to the fertilization system and the climate. Fertilization was conducted through organic manure and NPK using a fertigation system. Part of the organic fertilizer and NPK was applied before planting and the rest at flowering, a total of 35 tons/ha/year of manure was used. For chemical fertilizers, a composition of 80 kg/ha for nitrogen and 60 kg/ha for potassium and phosphate was applied. Approved phytosanitary products were used to prevent pest attacks.

#### 2.1.2. Brief Descriptions of the Cultivars Studied

The cultivars F1 Mongal and Petomech are among the most cultivated in Burkina Faso. They are consumed fresh or processed into puree or paste. They are also exported to neighboring countries, notably Ghana and Togo.

The Petomech cultivar is adapted to the cool and hot dry season. It is a plant with determined growth (height 1 m) and medium vigor, is very productive and needs to be staked. It is a semi-early cultivar, its earliness is of 75 days after transplanting. The fruits are elongated of bright red color and firm flesh, with an average weight of 90 g. It has intermediate resistance to *Verticillium* and *Fusarium* and is excellent for storage [26]. The average number of fruits per plant is estimated at 46 fruits. The yield per hectare is about 66.44 tons.

The cultivar F1 Mongal is an obtention of the National Institute of Agronomic Research (INRA French). It is an F1 cultivar, which can also be cultivated throughout the year. It is an early cultivar with a cultivation period of about 65 days, which gives round and slightly flattened fruits weighing about 100 g. It is a plant with determined growth, very good vigor and excellent fruit set. It has a high tolerance to bacterial wilt, resistance to TMV (0), to *Fusarium* FOL races 0 and 1, to *Fusarium* Forl and to *Stemphylium*. It is also resistant to root-knot nematodes (*Meloidogyne* spp.) [26]. Due to its excellent tolerance to bacterial wilt, it is more suitable for overwintering. Even under very high temperatures, the fruit set of this very robust cultivar allows it to produce many fruits.

### 2.2. Methods

#### 2.2.1. Seeds and Peels Collection

The seeds and peels were obtained from the collected tomato fruits. They were washed with tap water and distilled water. Fresh tomato fruits (3 kg) were collected from eight experimental fields in dry and rainy seasons each year for the three years of the study. The fruits were stored in the freezer at −20 °C. The peels were obtained by using a thermal shock. This method consists in immersing previously frozen tomato fruits in hot water at 90 °C for 30 s. The peel is easily detached without altering its nutritional value. These peels were then placed in a dryer at 40 °C for 96 h and then powdered. The seeds were obtained after the peeling of the fruits. They were collected and placed in a jar where water was added and kept in the open air at the temperature of the laboratory at 25 °C for 24 h. This technique allowed the digestion of the gelatin by the microscopic fungi and the seeds were released. These seeds were dried in a dryer and then powdered. The F1 Mongal seeds contained about 5.21% ± 0.2 of fresh weight whereas the peels were 10.5% ± 0.35%. In contrast, in the Petomech cultivar, the seeds contained only 3.4% ± 0.45 and the peels 12.78% ± 0.58 of the fresh weight. At least an amount of 200 g of powder of seeds or peels was available for all analyses.

#### 2.2.2. Analysis of Nutritional Parameters

##### Proximate Composition

The dry matter content was the weight of a sample obtained after drying in an oven at 103 °C ± 2 for 24 h. Moisture content was determined in accordance with AOAC method No. 943.06 (Section 31.1.10B), reported by Szabo et al. [27].

The determination of the pH of the tomato by-products was measured electro-metrically with a pH meter (PHS-25CW, Shanghai Benson Instrument Co., Ltd., Shanghai, China). Tomato peels or seeds powder (10 g) were put in a conical flask, then 100 mL of distilled water was added and the mixture homogenized until obtaining a homogeneous liquid. The samples were adjusted to a temperature between 22–25 °C. The pH meter was calibrated with buffer solutions as described in ISO-23,496 method [28].

The titratable acidity was determined by titration according to NF V05-101 (1974) standard [29]. A test sample of 10 g of by-product powder was weighed and put into a flask. Then 50 mL of distilled water was added and mixed until a homogeneous liquid was obtained. The contents of the flask were transferred to another 100 mL volumetric flask, adding distilled water up to the mark. After filtration, 10 mL of the filtrate was mixed with 10 mL of distilled water and few drops of phenolphthalein were added prior to titration with NaOH (0.1 M) until a persistent pink color was obtained for 30 s.

The total soluble carbohydrates was determined by the method of Dubois [30]. An aliquot of 100 mg ground test sample was mixed with 3 mL of 80% ethanol. The mixture was left at room temperature for 48 h, then ethanol was evaporated using a water bath at 100 °C, then 20 mL of distilled water was added to the dry residue. 4 mL of anthrone reagent was put into a test tube containing 2 mL of the obtained extract, and placed in a water bath at 62 °C for 8 min. After cooling in an ice bath, the tube was put to rest in the dark for 30 min, and the absorbance was read at 490 nm (Speectrophometer, Bio Tek instruments Inc., Winooski, VT, USA).

The protein content was determined according to the method of Bradford [31]. This method is based on a colorimetric assay detecting the color change of Coomassie blue at 595 nm, after complexation with aromatic amino acids (tryptophan, tyrosine and phenylalanine) and hydrophobic residues of amino acids present in the solution. The change in absorbance is proportional to the amount of dye bound, and thus to the concentration of protein in the sample. A standard curve was performed with a bovine serum albumin solution (Sigma-Aldrich, Merck KGaA, Darmstadt, Germany) ranging from 0 to 0.18 μg·μL^−1^.

Lipids quantification was performed by the Soxhlet extraction according to Lopez-Bacon [32]. The sample was placed in the cellulose cartridge and then in the Soxhlet apparatus. Then the flask was weighed and filled with hexane and the extractor was then mounted with a refrigerant. At least ten extraction cycles were operated. A test sample of 10 g of each by-product was used in triplicate to obtain the lipid content.

The potential energy values of the samples were calculated using Atwater coefficients according to the following formula: Energy value (kcal/100 g) = carbohydrate content (%) × 4 (kcal) + protein content (%) × 4 (kcal) + fat content (%) × 9 (kcal) [33].

#### 2.2.3. Determination of Ash Content

The ash content was determined according to the AFNOR method. For each sample, 5 g of ground powder was placed in a muffle furnace set at 550 ± 15 °C for 5 h until a grey, clear or whitish color was obtained [34]. The ash content was obtained by calculated mass difference.

#### 2.2.4. Determination of Mineral Content

Minerals (zinc, iron, magnesium) were quantified in triplicate by atomic absorption spectrometry (AAS) (PerkinElmer PinAAcle 900H, AZoNetwork UK Limited, Manchester, UK) according to ISO 1762 (2005) [35]. Specific instrumental parameters (lamp, wavelength, lamp current and slit width) were used for each mineral. Extraction was performed after incineration of 3 g of each sample in a muffle furnace at 550 °C for 10 h, followed by the dissolution of the ash with fuming hydrochloric acid (37%). Phosphorus was determined by continuous flow spectrometry (auto-analyzer) (Skalar Analytical B.V., Tinstraat 12, 4823 AA Breda, The Netherlands). Potassium and sodium were determined by flame photometry (Sherwood model). The standard solution concentration ranges were 0.5 mg/L, 2.5 mg/L, 5 mg/L, 7.5 mg/L and 10 mg/L for zinc and iron and 0.5 mg/L, 2 mg/L, 5 mg/L and 15 mg/L for magnesium, potassium and sodium. Highly concentrated samples were diluted prior to atomic absorption spectrometer analysis.

#### 2.2.5. Analysis of Proximal Phytonutrient Composition and Antioxidant Activity of Tomato By-Products

##### Levels of Phenolic Compounds

Polyphenols and flavonoids were extracted by maceration with 80% methanol with slight modification. The assay was performed using the Folin–Ciocalteu reagent with slight modification [36]. This method measures the intensity of the color of tungsten oxide, molybdenum oxide which is proportional to the amount of polyphenols. A concentration of 1 mg/mL was used for the spectrophotometric assay. All analyses were performed in triplicate and a calibration curve was obtained using gallic acid as standard.

The total flavonoid contents were determined by a spectrophotometric colorimetric method described by Arvouet-Grand et al. [37]. To 0.5 mL of sample with a mass concentration of 1 mg/mL, a solution of 2% AlCl_3_ was added. The total flavonoid content was calculated as quercetin equivalent from a calibration curve.

##### Determination of Lycopene and β-Carotene

The contents of β-carotene and lycopene were determined as previously described by Nagata and Yamashita [38]. The tests were performed in triplicate. The β-carotene and lycopene contents expressed as mg/100 g dry matter (DM). A test sample of 100 mg was dissolved in 10 mL of 70:30 (*v*/*v*) acetone-water and the reading was performed with a spectrophotometer at different wavelengths of 453 nm, 505 nm, 645 nm and 663 nm. The lycopene and β-carotene contents were calculated using the following formulas:Lycopene (mg/100 mL) = −0.0458·A663 + 0.204·A645 + 0.372·A505 − 0.0806·A453 
β-Carotene (mg/100 mL) = 0.216·A663 − 1.22·A645 − 0.304·A505 + 0.452·A453
where the letter A stands for the absorbance and the underscore numbers are the wavelengths.

##### Determination of Vitamin C

The method used for ascorbic acid quantification was based on its capacity of discoloration of 2, 6-dichlorophenolindophenol (DCPIP) [39]. The extracts (50 µL) were added to 150 µL of DCPIP (0.2 mM). Each test was performed in triplicate and the absorbance was read at 515 nm against a blank consisting of 150 µL DCPIP and 50 µL distilled water. A calibration curve was drawn with ascorbic acid at concentrations ranging from 10 µg/mL to 100 µg/mL. Vitamin C contents are expressed in mg ascorbic acid equivalent per 100 g dry matter (mg AAE/100 g DM).

##### Determination of Antioxidant Activity

Anti-free radical activity was assessed using DPPH, which was one of the first free radicals used to study the structure antioxidant activity relationship. The ability of the antioxidant to scavenge free radicals was evaluated by the percentage of DPPH discolorations in methanolic extracts at 1 mg/mL in triplicate [40]. The 2, 2′-diphenyl-1-picrylhydrazyl method was based on a spectrophotometric measurement of the changes in concentration of the DPPH radical resulting from its reaction with an antioxidant. The ability to scavenge the DPPH radical was measured spectrophotometrically at 517 nm. The percentage of inhibition was estimated based on the discoloration.

The antioxidant activity was determined by the ferric reducing antioxidant power (FRAP) method [41]. The assay is based on the reduction of ferric ion (Fe^3+^) to ferrous ion (Fe^2+^) which is accompanied by the appearance of an intense blue coloration quantifiable at 700 nm. A methanolic extract of concentration 1 mg/mL was used for the assay. The concentration of reducing compounds in the extract was expressed in of µg ascorbic acid equivalent (AAE/100 mg DM).

### 2.3. Statistical Analysis

The data were analyzed using descriptive statistics with the Microsoft Excel software version 2018. A Fisher test at the 5% level was performed for mean comparison. Principal component analyses (PCA) were performed using XLSTAT BASIC 2016 software, Microsoft Excel^®^. For each parameter the measurements were repeated in triplicates to minimize bias.

## 3. Results and Discussion

### 3.1. Results

The physicochemical parameters were significantly different (Figure 1). The pH of the peels of F1 Mongal was 4.5 ± 0.83, whereas for Petomech it was 3.38 ± 0.16. However, the pH levels of F1 Mongal and Petomech seeds were not significantly different (pH 5.93–6.01).

As for titratable acidity levels, they were 4.34 ± 1.06% and 5.27 ± 0.18% for F1 Mongal and Petomech peels, respectively, and 4.99 ± 0.27% and 4.94 ± 0.51% for F1 Mongal and Petomech seeds, respectively.

The peels had the highest moisture content with values ranging from 85.62 ± 3.9% for peels from F1 Mongal peels to 83.68 ± 0.95% for Petomech. The moisture content levels in seeds were 14.32 ± 1.91% for F1 Mongal and 16.8 ± 3.43% for Petomech. There were no significant differences among peels or seeds. However, the values for peels were very significantly different from those for seeds.

The contents of proteins, carbohydrates and lipids (Figure 2) in F1 Mongal and Petomech peels were 14.2% and 11.71%; 43.98% and 58.75%; and 5.4% and 5.76%, respectively. For seeds, they were 15.18% and 15.43%; 15.43% and 32.62%; and 22.26% and 21.55% in F1 Mongal and Petomech, respectively.

In terms of potential energy value, seeds had the highest value (472.8 kcal/100 g) and the peels presented the lowest value 280.47 kcal/100 g.

Ash contents of seeds and peels were significantly different (Figure 3). For minerals, the peels showed the highest potassium content (807.9 ± 0.01 mg/100 g). The highest magnesium and zinc contents were found in seeds with respective values ranging from 198.04 ± 0.01 mg/100 g to 230.54 ± 0.1 mg/100 g and 2.69 ± 0.002 mg/100 g to 2.81 ± 0.003 mg/100 g. The contents of all minerals are statistically different between cultivars.

The phytonutrient analyses of the by-products (Figure 4) showed values of total polyphenols ranging from 75 ± 1.36 mg GAE/100 g DM to 234.36 ± 0.06 mg GAE/100 g. The lowest value was found in the seeds of Petomech and the highest in the peels of the same cultivar.

For flavonoids, the values varied from 12.12 ± 0.64 mg QE/100 g DM to 105.87 ± 0.23 mg QE/100 g. The seeds of F1 Mongal displayed the lowest contents (12.12 ± 0.64 mg QE/100 g), whereas the peels of Petomech were richer in flavonoids (105.87 ± 0.23 mg QE/100 g).

For lycopene, the by-products showed levels varying from 14 ± 0.15 mg/100 g to 101.46 ± 0.46 mg/100 g. Unlike seeds, the peels did not show significant differences.

The β-carotene contents ranged from 6.34 ± 1.17 mg/100 g to 31.11 ± 1.34 mg/100 g. It was only in the seeds that the difference was significant.

The levels of vitamin C varied significantly among the seeds, contrary to the peels (Figure 4). The values ranged from 8.41 ± 0.4 mg/100 g DM to 24.82 ± 2.99 mg/100 g, respectively, in the seeds and peels of Petomech.

Tomato by-products showed strong antioxidant activities. For DPPH radical scavenging, the inhibition percentages ranged from 38.03 ± 1.74% to 60.8 ± 0.58%. However, for the same by-product, the differences were not statistically different. For the iron reducing activity (FRAP), the contents ranged from 67.17 ± 2.41 µg AAE/100 g to 110.35 ± 10.51 µg AAE/100 g DM. No significant difference was found.

Principal component analysis (Figure 5) between cultivars showed that the parameters were mostly clustered around the F1 Mongal cultivar. However, carbohydrates, titratable acidity and energy value were more correlated with Petomech. These results showed that Petomech contains the highest levels of carbohydrates and titratable acidity and is more energetic. However, these contents were not significantly different from those of F1 Mongal. In addition, the parameters were more dispersed in the Petomech cultivar than in the F1 Mongal cultivar.

The hierarchical classification (Figure 6) shows a proximity between the dry matter and lipid content, and a proximity between the pH and protein content. In contrast, titratable acidity is more correlated with moisture and carbohydrates, which are both correlated each other. These correlations show that the lipid content is a function of the dry matter of the by-products and that proteins influence the pH. Moisture leads to fermentation of carbohydrates, which increases the level of organic acids in the matrix, thus increasing the titratable acidity. These correlations resulted in three main groups (Figure 7). The first group (in blue) includes pH, protein, dry matter, lipids and energy content of the by-products which are correlated with each other. The second group (in yellow) is composed of moisture and carbohydrates that are more or less related. The third group (in grey) is formed by titratable acidity which is correlated to carbohydrates.

A principal component analysis, according to the cultivar and the by-products studied (Figure 8), presents four groups of which two large groups were made up of peels on the one hand and seeds on the other hand, titratable acidity not being correlated according to the by-product studied. The results show that the seeds are richer in protein and pH, have the highest dry matter and lipid content and are more energetic. However, high carbohydrate and water contents were found in the peels.

In terms of minerals (Figure 9), there were strong correlations between sodium and potassium in connection to ash content. Zinc level was more closely related to magnesium, which itself was positively correlated with iron. A negative correlation was also found between zinc and potassium.

### 3.2. Discussion

Moisture content varied in F1 Mongal and Petomech peels and for seeds. The highest and the lowest moisture contents were found in F1 Mongal peels and seeds. The high moisture content of the tomato fruit could be linked with the moisture content of the peel. Tomato is a climacteric fruit with high moisture content vulnerable to biotic and abiotic stresses [42].

It appeared that tomato seeds were less acidic than the peels. The low pH of the by-products may improve their storage and protects them from bacterial proliferation. The variations in pH level corroborate those found elsewhere with pH values of 4.85, suggesting that a low pH ensures a good conservation of the samples [43].

As for titratable acidity, the values were lower than those obtained on peels of four tomato cultivars with contents from 5.6% ± 1.21 to 6.3% ± 0.7 [44]. These differences showed that the samples would have undergone slight fermentation during drying.

The energy contribution of carbohydrates is estimated between 50 and 55%, suggesting that these nutrients are the major energy components in this fruit [45]. The consumption of refined carbohydrates is accompanied by an increase in the glycemic index. Refined carbohydrates increase the risk of diabetes mellitus, obesity and hypertensive diseases [46]. Thus, proposing new sources of carbohydrates with a low glycemic index is a healthy alternative. These by-products are also sources of dietary fiber important for the metabolism. Fibers facilitate the intestinal transit and help to avoid metabolic diseases such as constipation. Depending on their glycemic index, the by-products may prevent obesity and diabetes mellitus by reducing these complications [9].

The lipid contents of tomato peels were similar to other results [47]. Fatty acids and their derivative such as cutins are synthesized by plants to regulate certain infections by pathogens and to limit evapotranspiration. They are also a source of energy for living organisms and are responsible for several biological functions such as reproductive hormones. They carry fat-soluble vitamins such as vitamins A, E, D and K. Although high intake of lipids may cause malnutrition, their insufficient intake can lead to growth disorders and an increased risk of chronic diseases [20]. Animal fats, due to their high content of saturated fats, are decried in favor of vegetable ones. Fortunately, tomato seed oil contains more than 70% of unsaturated fatty acids and is rich in essential fatty acids [16]. Consumption of this oil prevents not only malnutrition, but also chronic diseases responsible for many deaths in the world [21]. With a lipid content of 22% (*w*/*w*) in seed, they contribute to covering the needs in oils because for a balanced diet the energy contribution of lipids is 34 to 40%. Tomato oil also contains β-carotene (provitamin A), and alpha-tocopherol vitamin E. Because of their role as carriers of fat-soluble vitamins and other bioactive compounds, tomato by-product lipids are important and should be valorized.

With respect to protein content, found levels were lower than other findings [48]. This difference could be explained by the difference of the cultivars and the pedoclimatic conditions. For the protein content of the peels, the results show that the peels of F1 Mongal had the highest overall content compared to those of Petomech. These results confirmed some findings that suggested the F1 Mongal cultivar was the best adapted to growing conditions and to high yields [49]. According to these findings, differences can be justified by the processing and storage methods. With these levels of proteins, tomato by-products can help to supply the protein requirement. Proteins are macromolecules involved in many functions in metabolism in the body such as in structure, defense, hormones transport, catalysis, etc. [14]. Insufficient protein intake can have serious consequences.

With contents of 371 kcal/100 g and 469 kcal/100 g DM, the F1 Mongal and Petomech seeds have the lowest and highest energy values of the seeds, respectively. The high energy value of the seeds is due to their high lipid content. Since recommended energy values for fortification are between 50 to 200 kcal/kg/person/day for children depending on nutritional status and priority is given to natural products, tomato by-products, and notably seeds, can be used for this purpose [50]. Thus, these by-products are all potential sources of energy that could help in achieving food and nutrition security in developing countries.

The ash content of the seeds corroborates the proportion of 3.8% previously found [51]. For peels, the ash content is higher than other tomatoes (3%). Therefore, tomato peels are a good source of minerals. The content in ash depends on cultivar. Due to their diverse functions in the body’s metabolism, minerals are indispensable. Their levels in tomato by-products revealed significant differences. The minerals in the peels of tomatoes grown in Burkina Faso can be classified in decreasing order of content as follows: potassium > magnesium > sodium > iron > zinc. In contrast, the ranking for seeds is: magnesium > potassium > sodium > iron > zinc. Indeed, essential minerals for humans, such as magnesium, calcium, iron, copper, zinc, iodine, sodium and selenium, can be found in tomatoes [13]. The by-products studied are important sources for the prevention of malnutrition, particularly among children, in a context where around 50% of children worldwide suffer from vitamin and mineral deficiencies [52]. Iron deficiency can cause anaemia, whereas magnesium may be an important nutrient in preventing and treating sarcopenia in older adults. Zinc is involved in many biological functions and is considered a multifunctional trace element, due to its ability to be a cofactor of several enzymes and transcription factors. It plays a role in oxidative stress response, homeostasis, immune responses, DNA replication, DNA damage repair, cell cycle progression, apoptosis and aging. Zinc is necessary for protein and collagen synthesis, contributing to wound healing and healthy skin [53]. Iron and iodine are considered to be the most important minerals in human metabolism [13]. The WHO recommends a healthy and balanced diet where the Na/K ratio is low in order to avoid cardiovascular accidents. While food habits are more oriented to energetic foods, it is necessary to develop new healthy and balanced nutritional sources [5]. Tomato by-products presented a low Na/K ratio (0.12), indicating their nutritional importance to prevent cardiovascular diseases and hypertension. Even if several foods can provide minerals, vegetables are well known as principal sources of fundamental minerals [54]. In this logic, tomato peels and seeds, which contain high levels of minerals, can be recommended as good sources of minerals to be explored in the fight against malnutrition.

Total phenolic compound contents were lower than the values obtained in other tomatoes which range from 904 to 1340 mg GAE/100 g DM [55]. This difference can be governed by genetical difference among cultivars and pedoclimatic conditions. However, they are comparable to the values of 265 ± 0.41 mg GAE/100 g DM obtained from other investigations [56]. These values corroborate that tomato by-products are potential sources of phenolic compounds. Although polyphenols may have pro-oxidant activities, they have long been considered to be real antioxidants. They may prevent and treat diseases due to oxidative stress and may prevent malnutrition such as obesity and premature ageing [10]. Although there is no indicated dose of polyphenols, their consumption may be beneficial to prevent the onset of several diseases. In addition, endogenous food polyphenols may protect the food against oxidation, a major health problem, and microbial growth. Tomato by-products, which are rich in total phenolics, can be used to obtain functional foods at low cost.

Flavonoid content levels obtained from peels were higher than those obtained from other tomatoes with values of 94 to 95 mg RUE/100 g [57]. This may be related to the extraction method, cultivar and agropedological and climatic conditions. Flavonoids are phenolics compounds generally involved in the brown, red and blue colors of flowers and fruits. They have several beneficial effects on human health. Flavonoids are potent antioxidants and antimicrobials that can prevent infectious diseases that are common in undernourished children in poor health conditions. Due to their antioxidant activities, they may prevent and treat obesity, including reducing chronic inflammation and the deposition of “bad cholesterol” (LDL), and prevent many metabolic and other chronic diseases [10].

Carotenoids are pigments generally responsible for the red, orange, yellow and green colors of fruits, vegetables and flowers. They are very diversified and have many biological roles. Lycopene and β-carotene are among the main carotenoids in tomato by-products. They are all known as powerful antioxidants. Lycopene may prevent inflammation, pain and cardiovascular disease. It may also regulate blood cholesterol levels and protect against high blood pressure and coronary heart disease [11]. As the most common carotenoid found in the blood, its consumption is very important. Due to its antimicrobial, antioxidant and anti-inflammatory role, lycopene is thought to be effective in preventing the effects of malnutrition [11]. In this study, the levels (101.23 to 101.46 ± 0.46 mg/100 g) obtained in the peels confirmed that tomato by-products are significant sources of bioactive compounds. In addition to lycopene, tomato by-products are also rich in β-carotene, known as the best provitamin A. In the body, it is non-stoichiometrically converted into retinol (vitamin A) to contribute to the achievement of vitamin A intake. The retinol is involved in the improvement of vision, and protects the skin against sunlight and fat oxidation [58]. Thus, tomato by-products can be recommended to structures and organizations working to combat malnutrition. They are natural sources of β-carotene at a low cost and are therefore accessible to vulnerable groups.

In studied tomato by-products, the highest vitamin C contents were from 21.38 mg/100 g to 24.82 ± 2.99 mg/100 g DM. Vitamin C (ascorbic acid) is a hydrosoluble vitamin which is involved in many of the body’s functions. It participates in the defense against viral and bacterial infections, the protection of the lining of the blood vessels, the assimilation of iron and in healing. It is nutritionally important because it participates in the synthesis of hydroxy-proline and collagen, protects the body from ageing, strengthens the immune system and is a free radicals scavenger [59]. Vitamin C, known as an anti-ascorbic chemical is also essential for the proper functioning of the nervous system. Vitamin C is important in the fight against malnutrition because it contribute to the reduction of avitaminosis, the consequences of which are fastidious for the body.

One of the important effects of the consumption of tomato by-products is their antioxidant power. Tomato by-products inhibited the DPPH radical and the iron reducing power (FRAP). This high antioxidant capacity may be due to their content of antioxidant compounds such as phenolics, carotenoids and vitamin C [22]. Therefore, these by-products may prevent the body from oxidative stress and cardiovascular and cancerous diseases. They may prevent obesity, hypertension and diabetes and treat inflammation [25]. Tomato by-products could be considered as nutraceuticals that are required to prevent and treat malnutrition, mainly in a context of scarce food diversity.

## 4. Conclusions

Adequate nutrition is a fundamental right that requires protection, promotion, food security, good health and adequate care. It is the main cause of the physical, mental and psycho-affective growth of children and adults. Unfortunately, there are several causes of deficiencies that hamper the development of many people. Since sources of functional foods are becoming increasingly scarce to meet the needs of populations, the exploitation of new avenues seems essential. Fortunately, tomato by-products, widely available in many developing countries, are a good source of primary nutrients as well as phytonutrients or nutraceuticals. These tomato by-products have high contents of protein, lipids, carbohydrates and also phenolic compounds, carotenoids and vitamin C. With these nutritional properties, tomato by-products can contribute strongly to the reduction of malnutrition. Thus, organizations working to eradicate hunger and malnutrition in the world can include tomato by-products in their formulations and as functional foods. Nevertheless, further investigations are needed to assess the bioavailability of several tomato nutrients and bioactive compounds.

## Figures and Tables

**Figure 1 nutrients-14-02871-f001:**
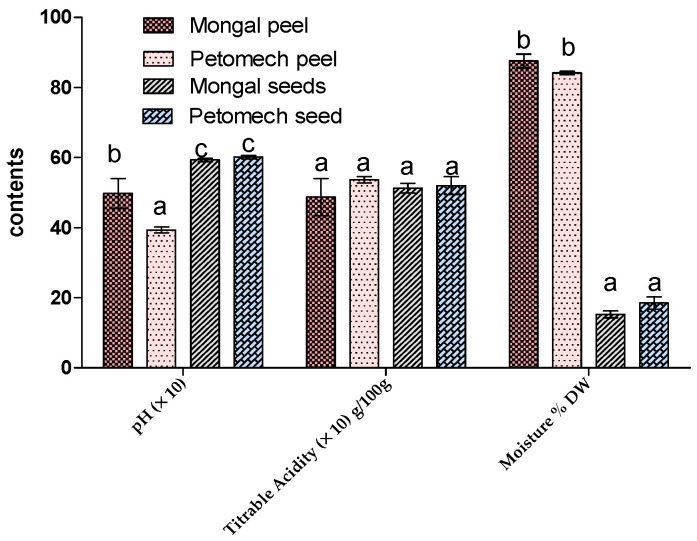
Physicochemical parameters of peels and seeds of Mongal and Petomech. The letters “a, b, c” indicate the different variations. Sticks with the same letters have statistically not different values. However, sticks with different letters have statistically different values.

**Figure 2 nutrients-14-02871-f002:**
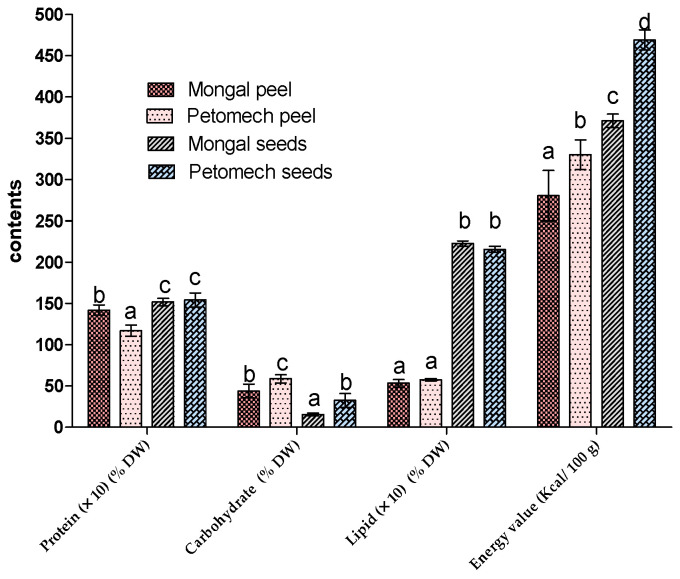
Biochemical composition of F1 Mongal and Petomech seeds and peels. The letters “a, b, c, d” indicate the different variations. Sticks with the same letters have statistically not different values. However, sticks with different letters have statistically different values.

**Figure 3 nutrients-14-02871-f003:**
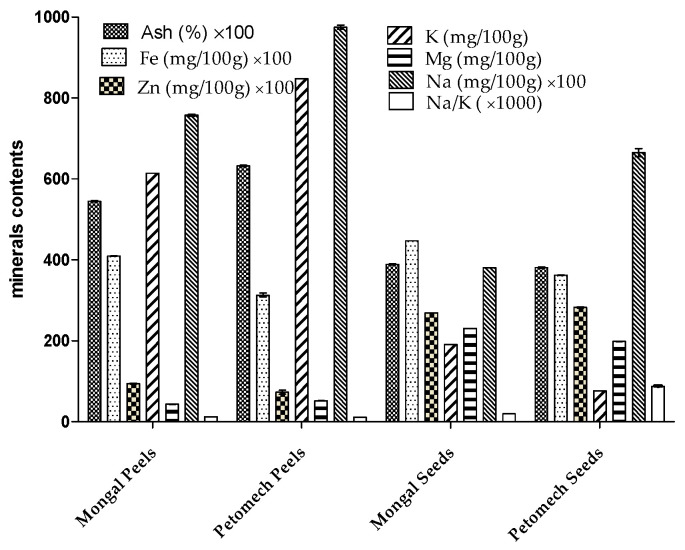
Ash and mineral contents of seeds and peels of F1 Mongal and Petomech.

**Figure 4 nutrients-14-02871-f004:**
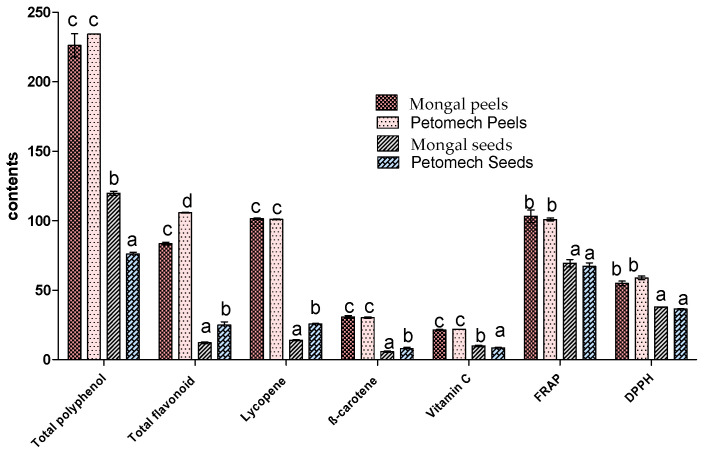
Proximal phytonutrient composition and antioxidant activity of tomato by-products. The letters “a, b, c, d” indicate the different variations. Sticks with the same letters have statistically not different values. However, sticks with different letters have statistically different values.

**Figure 5 nutrients-14-02871-f005:**
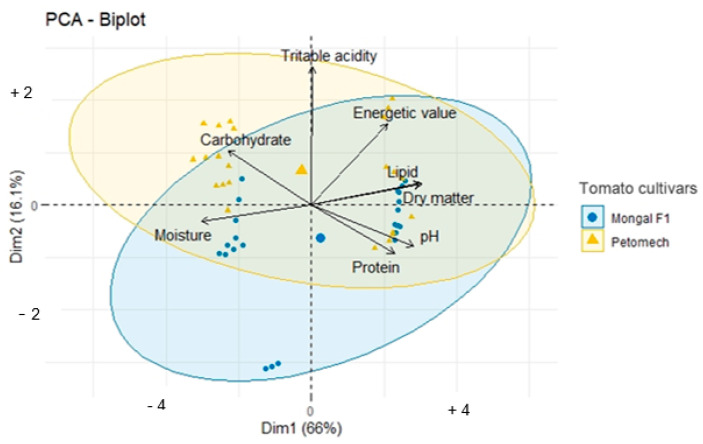
Principal component analysis between the two cultivars.

**Figure 6 nutrients-14-02871-f006:**
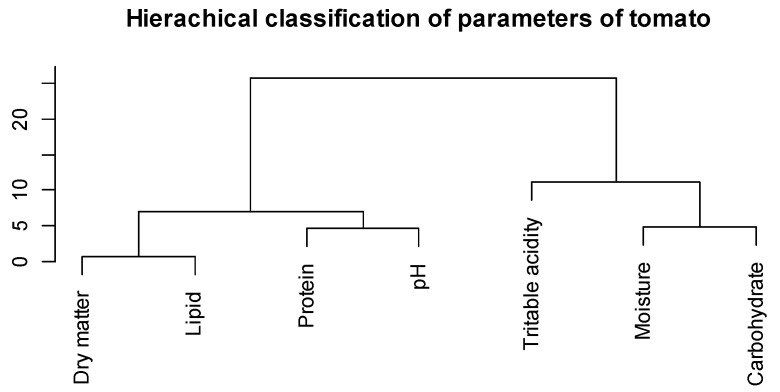
Hierarchical classification of tomato parameters.

**Figure 7 nutrients-14-02871-f007:**
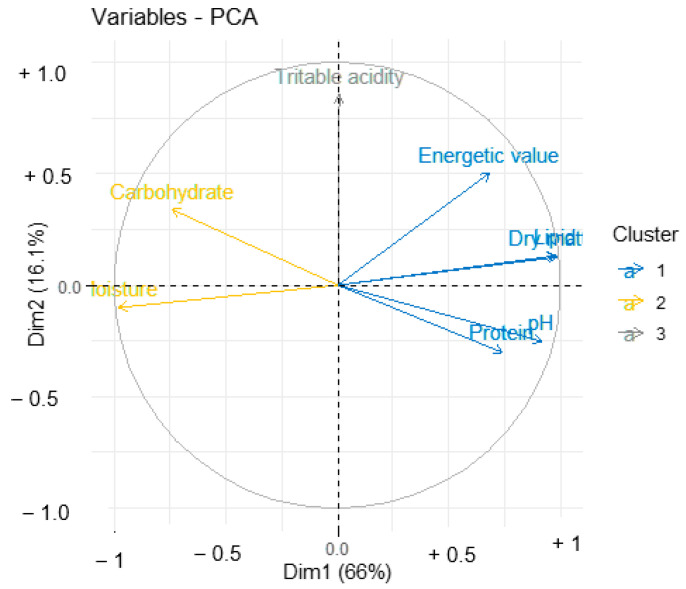
Principal component analysis of biochemical parameters.

**Figure 8 nutrients-14-02871-f008:**
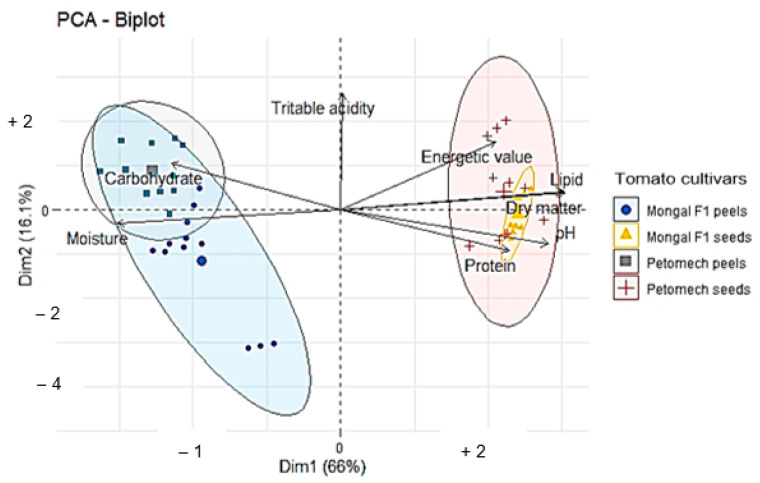
Biochemical parameters clustering according to cultivars.

**Figure 9 nutrients-14-02871-f009:**
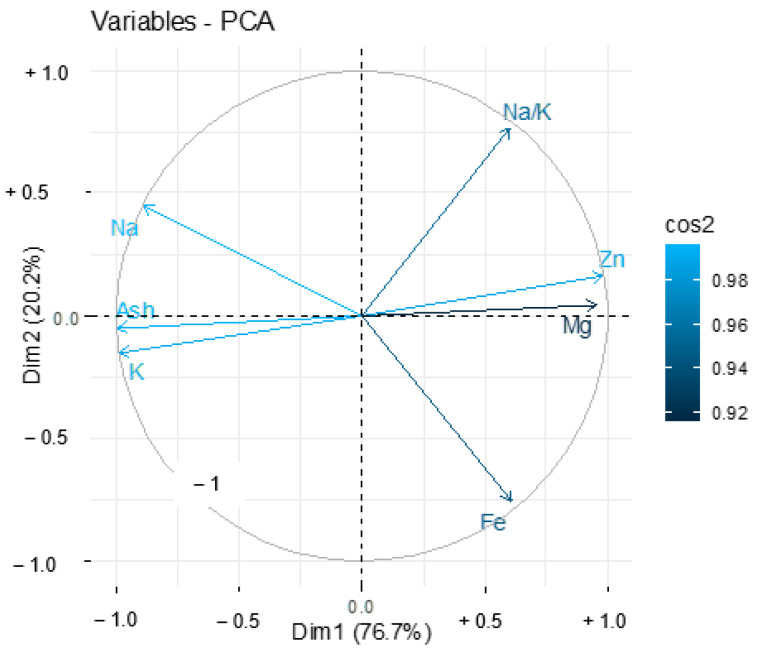
Correlation analysis of different tomato by-product minerals.

## Data Availability

Study data are available from the corresponding authors for researchers upon request.

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
