# Peer review of "Tomato By-Products, a Source of Nutrients for the Prevention and Reduction of Malnutrition"

_nutrients, 2022, doi:10.3390/nu14142871_

Round 1
Reviewer 1 Report
Dear Authors, here I have listed my comments.
1. Many comment of my rev-1 have been applied, however many inaccuracies have to be corrected;
2. Page 1, please, verify how to insert the Authors’ name. Download the template and use some recently published paper as a template. You do not have to write neither the name nor the first name completely in capital letters;
3. Page 1, authors’ name and authors contribution. Please, differentiate the two authors indicated as K.K.;
4. Page 1, authors name: first the first name and after the family name;
5. 2.1 sub-section, please, indicate the quantity of NPK and the quantity of manure p. Indicate also the period of fertilization and the procedure. The International reader wants to reproduce your experiment;
6. 2.1 sub-section, indicate the irrigation procedure: quantity;
7. Page 3, line 13: Verticillium, Fusarium in capital letter;
8. M&M section, describe the use of these two cv of tomatoes (fresh consumption? Juice?);
9. 2.2.1 sub-section, line 27 and in the whole manuscript: kg in small letter;
10. 2.2.2 sub-section, line 44 and in the whole manuscript, when you indicate a reference, do not use the system (by [2]), please write the author name and after [2], see recently published papers;
11. Page 4, line 11 and in the whole manuscript: 100 °C and not 100°C, separate;
12. Page 4, line 27 and in the whole manuscript, verify how to insert the references;
13. 3.1 sub-section and in the whole manuscript: when you indicate one figure or one table, use some recently published paper as a template (Figure 1 and not figure 1), please, verify;
14. The references section has to be arranged as per the guidelines of Nutrients. For example: the journal name abbreviated and italicized; the data of year, volume, pages are not correctly written. The issue number is not required. The punctuation is incorrect. Please, verify carefully because there is almost one mistake on each line. Be consistent with the guidelines of Nutrients and use some recently published paper as a template;
15. 2.1 sub-section (cultivar), Caption of figure 5 (variety), and also in your manuscript in relation with comments of your data, please, be consistent: cultivar or variety?
16. Please, when you make the numbering of the lines, do not re-start from each page but use the continuous condition;
17. Do not underline the correction and do not underline in the manuscript;
18. Please, apply carefully the instructions for Authors of Nutrients and evidence in blue color your corrections.
In my opinion, a major revision is necessary.
Regards.
Author Response
We thank the reviewer for concerns to improve the quality of the manuscript. Taking into account overall comment of the reviewer, the writing quality of the manuscript has been improved and drawn conclusions are supported by experimental data.
- Many comment of my rev-1 have been applied, however many inaccuracies have to be corrected;
Response:
On the first page, the abbreviation K. K._1 is used forand Kabakdé Kaboré and K. K._2 for Kiéssoun Konaté.
- Page 1, please, verify how to insert the Authors’ name. Download the template and use some recently published paper as a template. You do not have to write neither the name nor the first name completely in capital letters;
Response:
The presentation of the authors' names is reviewed using a recent nutrient article “Nutrients 2022, 14, 2344; https://doi.org/10.3390/nu14112344” according to your suggestion.
- Page 1, authors’ name and authors contribution. Please, differentiate the two authors indicated as K.K.;
Response:
On the first page, the K. K._2 stands for Kiéssoun Konaté.
- Page 1, authors name: first the first name and after the family name;
Response:
All authors' names have been rewritten starting with the first name and only the first letter is capitalized.
- 1 sub-section, please, indicate the quantity of NPK and the quantity of manure p. Indicate also the period of fertilization and the procedure. The International reader wants to reproduce your experiment;
Response:
We made a point of specifying the conditions under which these crops were grown, i.e. soil pH, type of fertilization, irrigation and quantities.
- 1 sub-section, indicate the irrigation procedure: quantity;
Response:
Irrigation and fertilization procedures and quantities are given in the materials and methods section.
- Page 3, line 13: Verticillium, Fusarium in capital letter;
Response:
We have taken this suggestion into account by capitalizing these two words and standardizing with the others.
- M&M section, describe the use of these two cv of tomatoes (fresh consumption? Juice?);
Response:
The consumption of the two cultivars is specified in the document.
- 2.1 sub-section, line 27 and in the whole manuscript: kg in small letter;
Response:
We have rewritten kg in lower case the whole document.
- 2.2 sub-section, line 44 and in the whole manuscript, when you indicate a reference, do not use the system (by [2]), please write the author name and after [2], see recently published papers;
Response:
The whole citation system has been revised.
- Page 4, line 11 and in the whole manuscript: 100 °C and not 100°C, separate;
Response:
Numerical values are separated from symbols throughout the document.
- Page 4, line 27 and in the whole manuscript, verify how to insert the references;
Responses:
The whole citation system has been revised.
- 1 sub-section and in the whole manuscript: when you indicate one figure or one table, use some recently published paper as a template (Figure 1 and not figure 1), please, verify;
Response:
The way in which the figures are quoted is revised.
- The references section has to be arranged as per the guidelines of Nutrients. For example: the journal name abbreviated and italicized; the data of year, volume, pages are not correctly written. The issue number is not required. The punctuation is incorrect. Please, verify carefully because there is almost one mistake on each line. Be consistent with the guidelines of Nutrients and use some recently published paper as a template;
Response:
References and citation style is reviewed using a recent nutrients article "Nutrients 2022, 14, 2344; https://doi.org/10.3390/nu14112344".
- 1 sub-section (cultivar), Caption of figure 5 (variety), and also in your manuscript in relation with comments of your data, please, be consistent: cultivar or variety?
Response:
In Figure 5 and 8, we have deleted variety in exchange for cultivar to be consistent with the text.
- Please, when you make the numbering of the lines, do not re-start from each page but use the continuous condition;
Response:
This suggestion is taken into account.
- Do not underline the correction and do not underline in the manuscript;
Response:
We used the correction model given by the Journal.
- Please, apply carefully the instructions for Authors of Nutrients and evidence in blue color your corrections.
Response:
The model given by the Journal suggests that the answers should be in red, so we have used red to respond to you.
In my opinion, a major revision is necessary.
We have taken all your suggestions into account. However, if you have made any omissions, please let us know so that we can take them into account.
Regards.
Reviewer 2 Report
This paper is fine, and a minor review is recommended. I have attached a pdf file with a few comments for the authors to improve.
Author Response
We thank you for your suggestions for improving the document. Taking into account overall comment of the reviewer, the writing quality of the manuscript has been improved and drawn conclusions are supported by experimental data.
In the abstract, we have separated the values of seeds and peels.
In the introduction, you suggested that we relativize the state of malnutrition because not all countries experience the same realities. We have taken this into account.
In the section on materials and methods, you proposed rewording that we have taken into account.
In the results section, we have removed the title of the axes in accordance with your suggestion.
Reviewer 3 Report
The study deals with the quality of peel and seeds as tomato by-product with regard to possibility of using it in the reduction of malnutrition. Two cultivars were tested. The results could be useful, but the work is not entirely clear, sampling for analysis is unclear. It is not known what influenced the obtained results. The way the results are presented is not friendly to the reader. Some of the statements are speculative.
Introduction - it is very poor in the context of the issue explored. It is basically limited to a commentary regarding one ingredient. On the other hand, the aspect of malnutrition is over-exploited. I understand, the issue of malnutrition is an important problem (and it currently may get worse), but authors should focus on the studied material in relation to other research, and it would be valuable to show some novelty too.
Mat and Met. The authors write in the materials and methods that the research was conducted in 3 years, samples were derived from 4 regions, but what does it really mean? In the presented results there is no reference to the growing season or the region. How were samples collected for analysis? What consisted of one repetition? How many grams of peel and seeds can be obtained from, for example, one kilogram of tomatoes. The soil and weather conditions are described very generally. There is no detailed data on the pH of the soil, the content of organic matter and the content of essential nutrients. The authors themselves comment that these are factors that significantly influence the chemical composition.
The conclusions go beyond the scope of the conducted research.
Detailed remarks.
L. 23. ‘f’ before 5.4 – what does it mean? Presenting results authors should reconsider the number of significant figures and avoid exaggerated accuracy.
Key words: Three of the five keywords are already in the title, what as a rule should be avoided.
L. 42 . What two forms of malnutrition are we talking about here - it is better to avoid such forms as 'these', 'those' etc., please call them exactly.
L. 130/131. Unclear, unfinished sentence.
L. 151. What does it ‘0.1N’ mean? please use SI unit.
L. 103/117. Reference is missing.
L.166. The reference cited here has number 33, while the previous one is 23, which is a jump of 10 positions.
L. 190. Please add the method used. Diverse description of the analytical methods used. For some, the descriptions are very short, for others - lengthy. The same principle should be used in the description for standard method (weight of sample tissue, extractor, analytical technique …e.t.c. and reference which describes a given method) unless the method used is new/innovative and then needs more details.
L. 207. EAA - no explanation of the abbreviation.
Graphs - Multiple data in one graph: different components (with different units); different varieties, types of tissues tested. No legend. What's more, the style used (the colors of the columns) - very difficult to read. In my opinion, Tables with averages +/- SD would be a better form.
Author Response
Response to Reviewer 3 Comments
We acknowledge performed qualitative corrections in order to improve the manuscript. Taking into account overall comment of the reviewer, the writing quality of the manuscript has been improved and drawn conclusions are supported by experimental data.
The study deals with the quality of peel and seeds as tomato by-product with regard to possibility of using it in the reduction of malnutrition. Two cultivars were tested. The results could be useful, but the work is not entirely clear, sampling for analysis is unclear. It is not known what influenced the obtained results. The way the results are presented is not friendly to the reader. Some of the statements are speculative.
Responses :
We have considered this concern by improving the materials and methods section. We have also revised some figures to be consistent with the legend of other figures and with the text of the main document.
Introduction - it is very poor in the context of the issue explored. It is basically limited to a commentary regarding one ingredient. On the other hand, the aspect of malnutrition is over-exploited. I understand, the issue of malnutrition is an important problem (and it currently may get worse), but authors should focus on the studied material in relation to other research, and it would be valuable to show some novelty too.
Response :
In response to these observations, we have deepened the introduction by discussing other nutrients such as phenolics, lycopene, β-carotene, minerals and other macronutrients like carbohydrates and proteins.
Mat and Met. The authors write in the materials and methods that the research was conducted in 3 years, samples were derived from 4 regions, but what does it really mean? In the presented results there is no reference to the growing season or the region. How were samples collected for analysis? What consisted of one repetition? How many grams of peel and seeds can be obtained from, for example, one kilogram of tomatoes. The soil and weather conditions are described very generally. There is no detailed data on the pH of the soil, the content of organic matter and the content of essential nutrients. The authors themselves comment that these are factors that significantly influence the chemical composition.
Response :
In the material and methods section, we did not take into account the production regions. In fact, this study focused on the average levels in three rainy seasons of crops with similar pedoclimatic conditions and the same treatments. The study did not only aim to compare the levels according to the areas of cultivation but it gives a view on the average levels of nutrients. In addition, we made a point of specifying the conditions under which these crops were grown, i.e. soil pH, type of fertilization, irrigation and pluviosity. We also specified the quantities of used seeds and peels for the studies. We discussed that the differences may be related to soil and climatic conditions. Furthermore, cultivars under the same conditions do not have the same adaptive systems
The conclusions go beyond the scope of the conducted research.
Response :
We have revised the conclusion to take into account this concern
Detailed remarks.
- 23. ‘f’ before 5.4 – what does it mean? Presenting results authors should reconsider the number of significant figures and avoid exaggerated accuracy.
Response :
The way the results are presented is revised to meet your suggestion. The ''f'' is an error, we have removed it.
Key words: Three of the five keywords are already in the title, what as a rule should be avoided.
Response :
The keywords have been improved..
- 42 . What two forms of malnutrition are we talking about here - it is better to avoid such forms as 'these', 'those' etc., please call them exactly.
Response :
We have specified the two forms of malnutrition that we have mentioned to avoid any ambiguity.
- 130/131. Unclear, unfinished sentence.
Response :
The sentence has been rewritten and the punctuation respected.
- 151. What does it ‘0.1N’ mean? please use SI unit.
Response :
"0.1N" is 0.1 mol/L, we have specified this.
- 103/117. Reference is missing.
Response :
References and citation style is reviewed using a recent nutrients article "Nutrients 2022, 14, 2344; https://doi.org/10.3390/nu14112344".
L.166. The reference cited here has number 33, while the previous one is 23, which is a jump of 10 positions.
Response :
The reference has moved from 23 to 33 due to the addition of new references.
- 190. Please add the method used. Diverse description of the analytical methods used. For some, the descriptions are very short, for others - lengthy. The same principle should be used in the description for standard method (weight of sample tissue, extractor, analytical technique …e.t.c. and reference which describes a given method) unless the method used is new/innovative and then needs more details.
Response :
The methods have been reviewed; the essence of each method is provided in accordance referee’s suggestion.
- 207. EAA - no explanation of the abbreviation.
Response :
EAA: ascorbic acid equivalent, we made a point of writing it out entirely for the first appearance.
Graphs - Multiple data in one graph: different components (with different units); different varieties, types of tissues tested. No legend. What's more, the style used (the colors of the columns) - very difficult to read. In my opinion, Tables with averages +/- SD would be a better form.
Responses :
For the figures, we have reworded the legends so that all the figures have the same legends in order to make them easier to read. You will see the red sticks representing the peels in all the graphs and the grey or even dark colours for the seeds. We have preferred the figures to the tables because we think that the differences are more perceptible with the figures.
Round 2
Reviewer 1 Report
The Authors have included all my comments. The argument
is now well presented, the experiment is properly designed and data are well discussed.
I suggest the publication of this manuscript in the present form.
Regards.
This manuscript is a resubmission of an earlier submission. The following is a list of the peer review reports and author responses from that submission.
Round 1
Reviewer 1 Report
Dear Authors, here I have listed my comments.
a. The argument is interesting but the manuscript has to be implemented and better presented. The introduction section has to better describe the state of the art. The M&M section has to be extended and completed. The references section has to be arranged as per Nutrients guidelines. Inaccuracies in the text.
b. Page 1, re-arrange the affiliation of each author, verify spacing between words and be consistent with the template;
c. Page 1, verify how to insert the names of authors, be consistent with the template and use some recent published paper as a sample;
d. Key words in English: replace sous-produits with by-products;
e. Introduction section, indicate the quantity of tomato produced in the world and in your Country and support this statement with some International reference, for example an International website, I suggest two, but you ca include more (1-2):
(1) https://www.statista.com/statistics/1030637/us-market-tomato-production-value/
(2)ttps://www.statista.com/statistics/264065/global-production-of-vegetables-by-type/
f. Introduction section, lines 48-52. The nutrients contained in the tomato by-products have to be described. One of the most important by-products is the tomato seed oil and there is a lot of studies regarding to the TSO composition, whereas you have included only two references not representing the actual state of the art of this resource. Please, find, read and discuss about findings on Tomato Seed Oil related to: the effect of cultivar (3); The cold break, hot break pre-treatments and the harvest year effect (3); The effect of the extraction system (5); the fatty acid composition (3-4-5), the sterol composition (6); the policosanol content (7) and the n-alkane composition (8):
3.Physicochemical composition of tomato seed oil for an edible use: the effect of cultivar.
Int. Food Res. J. 23(2): 583-591 (2016).
4.Tomato seed oil for edible use: cold break, hot break and harvest year effects.
Journal of Food Processing and Preservation 2017; 41(12):e13309. https://doi.org/10.1111/jfpp.13309
5.Tomato seed oil: a comparison of extraction systems and solvents on its biodiesel and edible properties.
Riv. Ital. Sostanze Gr. 94 (3) 149-160 (2017).
6.Sterol composition of tomato (Solanum lycopersicum L.) seed oil: the effect of cultivar.
Int. Food Res. J. 23 (1) 116-122 (2016).
7.Policosanol in Tomato Seed Oil (Solanum lycopersicum L.)
8.n-Alkanes in tomato seed oil (Solanum lycopersicum L.)
g. Introduction section, when you discuss about tomato seed and tomato seed oil, describe the beneficial effect of fatty acids (MUFAs, PUFAs, EFAs) (9); sterols (10-11) and policosanol (12-13) on the human health and support this discussion with proper references:
9.Beneficial effects and oxidative stability of omega-3 long-chain polyunsaturated fatty acids.
Trends in Food Science & Technology 25, Issue 1, May 2012, Pages 24-33.
https://doi.org/10.1016/j.tifs.2011.12.002
10.Serum sterol responses to increasing plant sterol intake from natural foods in the Mediterranean diet.
Eur J Nutr (2009) 48:373–382 DOI 10.1007/s00394-009-0024-z
11. Plant sterol consumption frequency affects plasma lipid levels and cholesterol kinetics in humans.
European Journal of Clinical Nutrition (2009) 63, 747–755.
12.Policosanols: Chemistry, Occurrence, and Health Effects
Current Pharmacology Reports (2019).
DOI: 10.1007/s40495-019-00174-9
13. Taylor JC, Rapport L, Lockwood GB. 2003. Octacosanol in human health. Nutrition 19, 192–195.
h. 2.1 sub-section: no information is given about the agronomic condition for tomato production: type of soil; fertilizers (type, quantity, period); irrigation (type, quantity, period); description of plants and tomatoes (biometrics including peel and seeds content);
i. M&M section, describe the use of these two cv of tomatoes;
j. 2.1 sub-section, the year of the experiment is missed;
k. 2.2.1 sub-section: how many tomatoes were used in this experiment? From how many plants? (Biological sample);
l. 2.2.1 sub-section and in the whole manuscript, when you indicate a temperature, separate the numeric value from the symbol: -20 °C and not -20°C;
m. 2.2.2 sub-section, line 81, verify the spacing before the first bracket and insert a dot at the end of the sentence;
n. Line 88, verify the indent at the begin of the sentence;
o. Line 95: -1 as an exponent,
p. 2.2.3 sub-section, line 100 and in the whole manuscript, separate the numeric value from the unit: 5 g and not 5g;
q. Line 94 µl, line 109 mL, please be consistent in the whole manuscript: small or capital letter for liter? I suggest always L;
r. 3.1 sub-section, line 119 and in the whole manuscript: when you indicate one figure or one table, use some recently published paper as a template;
s. Line 120, delete one space after …. of and before 4.5;
t. Line 134, insert one space before Kcal;
u. Figure 1 and 2, the statistical analysis is missed. Only SD?
v. Line 188, delete one space before 90%;
w. Line 190, delete one space before …. Tomato;
x. 3.2 section, line 205, do not begin the sentence with the reference number, do not use [27] showed. The same in the lines 200-201, 211-212, 237, 268 and in the whole manuscript: re-arrange and use some recent published paper of Nutrients as a sample;
y. The references section has to be arranged as per the guidelines of Nutrients;
z. Please, evidence your corrections.
In my opinion, a major revision is necessary.
Regards.
Reviewer 2 Report
Dear Authors,
the peer-reviewed article concerns with two topics that are very important for the modern world: the first is the use of waste from the tomato processing industry and the second is the prevention and reduction of malnutrition. The authors in their research show that tomato by-products are of great nutritional and technological importance in the formulation of various food products for consumption, which may contribute to the fight against malnutrition.
However, the presented subject is not new. This topic has been explored by researchers all over the world for years, as evidenced by a huge list of publications (e.g Szabo, K., Cătoi, AF. & Vodnar, D.C. Bioactive Compounds Extracted from Tomato Processing by-Products as a Source of Valuable Nutrients. Plant Foods Hum Nutr 73, 268–277 (2018). https://doi.org/10.1007/s11130-018-0691-0; Silva, Yasmini & Borba, Bárbara & Pereira, Vanessa & Reis, Marcela & Caliari, Márcio & Brooks, Marianne & Ferreira, Tânia. (2018). Characterization of tomato processing by-product for use as a potential functional food ingredient: nutritional composition, antioxidant activity and bioactive compounds. International Journal of Food Sciences and Nutrition. 70. 1-11. 10.1080/09637486.2018.1489530); Maya Ibrahim, Madona Labaki, CHAPTER THREE - Extraction and formulation of valuable components from tomato processing by-products Editor(s): Mejdi Jeguirim, Antonis Zorpas, Tomato Processing by-Products, Academic Press, 2022, Pages 77-116, ISBN 9780128228661, https://doi.org/10.1016/B978-0-12-822866-1.00009-0).
The presented research does not add any new original information. They are very general. The authors measure the protein content, while other researchers go deeper and report the amino acid composition of these proteins, the peer-reviewed article presents the fat content, and other articles can find the fatty acid composition, etc. Important ingredients such as calcium, phosphorus, copper, carotene or phenolic compounds responsible for antioxidant properties are omitted from the nutritional values.
Below are some minor ambiguities to which the reviewer has comments:
line 21 - What does "energetic macronutrients" mean?
line 30 - I suggest to delete "sous-produits" and replace it with another statement
line 41 - "... both forms of malnutrition ..." proposes to clearly indicate these two forms
line 79 - please specify the drying time and temperature
line 81 - useless space before [11] and missing dot after
lines 82-83 - please describe how to prepare the sample for pH measurement
line 86 - no dot at the end of the sentence
line 87 - please briefly describe the principle of the method
line 97 - there is (Kcal/100g), it should be (Kcal/100 g)
line 100 - there is 5 g, it should be 5 g
line 102 - inappropriate form of citation; no citation in the bibliography
lines 107 -108 - "Potassium and sodium was determined by flame photometry (Sherwood model)" and the other minerals how were they marked? Please provide more details of the AAS measurement
lines 109-110 - "... 0.5 mg / L, 2 mg / L, 5 mg / L, 15 mg / L respectively for magnesium, potassium and sodium." - 4 ranges are given but only 3 elements
lines 114 - 116 - please specify the number of repetitions for each measurement - it may be in chapter 2.3. Statistical analysis or each measurement method separately
lines 119 - 139 - please add the name of the sample (e.g. Petomech peel) to the results, because in Figure 1 the order of samples is different than in Figure 2 and it is quite confusing
line 119 - is (Figure n ° 1), it should be (Figure 1)
line 127 - it's 14.32 ± 1.91 should be 14.32 ± 1.91%
lines 130 - 139 - Are contents of proteins, carbohydrates and lipids and minerals on a dry weight basis?
line 130 is (Figure n ° 2), it should be (Figure 2)
line 133 - there is (472.8 Kcal / 100g), it should be (472.8 Kcal / 100 g)
line 134 - it's 280.47 Kcal / 100g should be 280.47 Kcal / 100 g
lines 137-138 - no units
line 139 - no dot at the end of the sentence
Figure 1 - reverse the order of samples in the legend as in Figure 2; no units in the figure
figure 2 - no units; in the axis signature there should be Energy value and not Energetic value
Figure 3 - please explain the abbreviations under the graph (MP, PP, MS, PS)
lines 185 - 201 - the values of dry weight and moisture content should add up to 100% and this is not the case here, for example for Mongal peels: moisture = 85.62% and dry matter = 9.69%. Were dry weight and moisture content determined as two separate measurements? Was one derived from the other?
lines 202 - 203 - other pH values are given in section 3.1. Results
line 234 - "for seeds" is missing at the end of the sentence
lines 236 - 237 - see line 234
line 248 - is "alpha linoleic and linoleic acid" should be "alpha linolenic and linoleic acid"
line 262 - there is "petomech peels" should be "Petomech peels"
lines 273-274 "The protein content is a good indicator because at these levels, these by-products are able to form absorption buffers that facilitate digestion [53]." - indicators of what? Please rewrite the sentence
lines 290-291 - "The fortification recommended energy values are between 100 Kcal / ml, and natural products are prioritized "- enriching what, who?
line 295 - is "Peels" should be "peels"
References - no year
Best regards,